# Event-Triggered Bounded Consensus Tracking for Second-Order Nonlinear Multi-Agent Systems with Uncertainties

**DOI:** 10.3390/e25091335

**Published:** 2023-09-15

**Authors:** Ying Ma, Chan Gu, Yungang Liu, Linzhen Yu, Wei Tang

**Affiliations:** 1School of Electrical and Control Engineering, Shaanxi University of Science and Technology, Xi’an 710016, China; 210611016@sust.edu.cn (Y.M.); tangwei@sust.edu.cn (W.T.); 2School of Control Science and Engineeringhl, Shandong University, Jinan 250100, China; lygfr@sdu.edu.cn (Y.L.); yulz@maill.sdu.edu.cn (L.Y.)

**Keywords:** nonlinear multi-agent systems with uncertainties, bounded consensus tracking, adaptive control, triggering mechanism

## Abstract

This paper is concerned with event-triggered bounded consensus tracking for a class of second-order nonlinear multi-agent systems with uncertainties (MASs). Remarkably, the considered MASs allow multiple uncertainties, including unknown control coefficients, parameterized unknown nonlinearities, uncertain external disturbances, and the leader’s control input being unknown. In this context, a new estimate-based adaptive control protocol with a triggering mechanism is proposed. We rule out Zeno behavior by testifying that the lower bound on the interval between two consecutive events is positive. It is shown that under the designed protocol, all signals caused by the closed-loop systems are bounded globally uniformly and tracking errors ultimately converge to a bounded set. The effectiveness of the devised control protocol is demonstrated through a simulation example.

## 1. Introduction

Nowadays, consensus tracking has been extensively investigated for MASs due to its widespread applications in many fields such as cooperative control of mobile robots and spacecraft formation flying control [1,2,3,4,5,6]. In MASs, each agent is usually equipped with embedded microprocessors, which have limited energy and computing resources. We remark that event-triggered control (ETC) makes it so the information transmitted or the controller is updated only when necessary for systems, which thus can effectively reduce resource consumption. As an effective tool for reducing resource consumption, event-triggered control has been proposed [7,8]. This inspires the study of the event-triggered consensus tracking problem of MASs.

In the past few decades, abundant work on consensus tracking of MASs with ETC has been reported [9,10,11,12,13,14,15,16,17,18,19]. Specifically, single-integrator agents and double-integrator agents were investigated in [9,10], respectively. However, the presented triggering mechanism required each agent to continuously monitor the states of neighboring agents. This requirement is removed in [11] by adopting a triggering threshold that is state-independent, while consensus tracking of general linear MASs was realized. As a further improvement, works [12,13] considered the consensus tracking problems by ETC for nonlinear MASs.

We remark that all the above works do not consider MASs with uncertainties, which are unavoidable when modeling real plants. Specifically, work [20] proposed a hybrid system approach to address the ETC problem for linear systems with uncertainties. Then, in [21], consensus tracking was realized for nonlinear MASs whose the parameters in nonlinearities have a known upper bound. As an extension, work [22] permitted completely unknown parameters in nonlinearities. Work [23] considered MASs with parameterized unknown nonlinearities, but unknown control coefficients were not taken into account. Work [24] considered uncertain nonlinear MASs with unknown control coefficients, and an estimate-based ETC protocol was proposed to achieve bounded consensus tracking. Then, uncertain nonlinear MASs subjected to unknown external disturbances were investigated in [25], and output consensus tracking was realized. However, in reality, second-order nonlinear MASs with uncertainties are more practical for modeling real systems such as torque motors and jet engines, which are tuned to achieve the desired motion directly through acceleration rather than velocity [26]. Hence, it is exigent to investigate consensus-tracking-based ETC for nonlinear MASs with uncertainties. In this context, how to design an ETC protocol to realize bounded consensus tracking and how to develop an appropriate compensation mechanism to counteract the uncertainties in systems deserve our efforts. A comparison of the features of the investigated MASs in the existing literature is depicted in Table 1.

This paper is devoted to event-triggered bounded consensus tracking for second-order nonlinear MASs with uncertainties. Compared with the relevant literature, the MASs in this paper permit multiple uncertainties, including unknown control coefficients and parameterized unknown nonlinearities. In contrast with work [27], the considered nonlinear MASs with uncertainties permit unknown external disturbances, and event-triggered consensus tracking is investigated in this paper. Moreover, the leader’s control input is also allowed to be unknown, and the leader’s information is broadcast to only a few of the agents. To counteract the uncertainties and to realize bounded consensus tracking, a novel estimate-based adaptive control protocol with triggering mechanism is developed. Further, through Lyapunov analysis, it is proved that the devised adaptive control protocol can make certain that all signals caused by the closed-loop systems are bounded globally uniformly, and with the passage of time, tracking errors converge to a bounded set.

The remaining context of this paper consists of the following sections. The related preliminaries, which include the problem statement, graph theory, and notation, are formulated in Section 2. Section 3 proposes the triggering mechanism and a distributed adaptive control protocol with the triggering mechanism. The main results are summarized in Section 4. Simulations of nonlinear MASs with uncertainties are given to exemplify the effectiveness of the proposed control protocol are in Section 5. Concluding remarks are provided in Section 6.

## 2. Preliminaries

### 2.1. Graph Theory and Notation

Represent a graph by G=(V,E), which consists of a nonempty finite set of nodes and a set of edges E⊆V×V, where a set of nodes is denoted by V = ($v_{1}$, …,$v_{N}$). The edge ($v_{i}$,$v_{j}$) denotes that node $v_{i}$ can transmit information to node $v_{j}$, where $v_{i}$ is $v_{j}$’s neighbor. Graph G is called undirected if ($v_{i}$,$v_{j}$) ∈E implies ($v_{j}$,$v_{i}$) ∈E. For a directed graph, an edge ($v_{i}$,$v_{j}$)∈E indicates that node $v_{i}$ can transmit information to node $v_{j}$, but node $v_{j}$ cannot transmit information to node $v_{i}$. A directed graph contains a spanning tree: the graph includes a root node with no parent and the root node has directed paths to every other node.

The adjacency matrix of a graph G is denoted by A = [$a_{ij}$] ∈ $R^{N\times N}$. In A, $a_{ij}$ = 1 if ($v_{i}$,$v_{j}$)∈E, and $a_{ij}$ = 0 otherwise. This paper assumes that there are no self-loops; thus, $a_{ii}$ = 0. Correspondingly, the degree matrix is denoted by Δ = diag{$\Delta_{1},\cdots ,\Delta_{N}$} with $\Delta_{i}$ = $\sum_{j=1}^{N}$$a_{ij}$. Then, the Laplacian matrix associated with G can be defined as L = Δ−A, where L = [$l_{ij}$] ∈ $R^{N\times N}$ is defined as $l_{ii}$ = $\sum_{j=1,i\neq j}^{N}$$a_{ij}$, and $l_{ij}$ = $-a_{ij}$, i≠j.

In this paper, we use $\mu_{i}$ = 1 to denote that the leader’s trajectory information is available for the *i*-th follower; otherwise, $\mu_{i}$ is set as 0. Let $P=\mathrm{diag}\{\mu_{1},\cdots ,\mu_{N}$}, and define *W* = L + P.

Standard notations are presented as follows.
*Q* > 0the matrix *Q* is positive definite$Q^{T}$the transposition of matrix *Q*‖Q‖the 2-norm of a matrix *Q*$\lambda_{\max}\left(\cdot \right)$the largest eigenvalues of the matrix$\lambda_{\min}\left(\cdot \right)$the smallest eigenvalues of the matrix$1_{N}$an *N*-dimensional column vector with all entries being 1$D^{+}$the upper-right-hand derivative

**Lemma** **1.**
*If a directed graph contains a spanning tree wherein the root node is the leader, then the matrix W = L + P is positive definite.*


**Lemma** **2** (Schur Complement).
*For a symmetric matrix Q = $\left[\begin{array}{cc} Q_{11} & Q_{12}\\ Q_{12}^{T} & Q_{22} \end{array}\right]$, the following statements are equivalent:*
*(1)* 
*Q>0,*
*(2)* *$Q_{11}>$*0, *$Q_{22}-Q_{12}^{T}Q_{11}^{-1}Q_{12}>$*0,*(3)* *$Q_{22}>$*0, *$Q_{11}-Q_{12}Q_{22}^{-1}Q_{12}^{T}>$*0.


### 2.2. Problem Statement

Consider an uncertain MAS with *N* followers and one leader. The *i*-th follower’s dynamics, i=1,⋯,N, are described as follows:(1)$$\left\{\;\;\;\begin{array}{cc} & {\dot{x}}_{i}\left(t\right)=v_{i}\left(t\right),\\ & {\dot{v}}_{i}\left(t\right)=b_{i}u_{i}\left(t\right)+\theta_{i}f_{i}\left(x_{i},v_{i},t\right)+d_{i}\left(t\right),\;i=1,\cdots ,N, \end{array}\right.$$
where $x_{i}\left(t\right)$∈ R and $v_{i}\left(t\right)$∈ R represent the *i*-th follower’s states; $u_{i}$∈ R is the *i*-th follower’s control input; $b_{i}$∈ R is a non-zero constant that is unknown, called the *i*-th follower’s control coefficient; $\theta_{i}$∈ R is an unknown constant; $f_{i}\left(x_{i},v_{i},t\right):R\times R\times R_{+}\to R$, is a known nonlinear function and is locally Lipschitz in ($x_{i},v_{i}$) on R×R and continuous in *t* on $R_{+}$; $d_{i}\left(t\right)$∈ R represents an external disturbance that is unknown and time-varying.

The leader’s dynamics are described by:(2)$$\left\{\;\;\;\;\begin{array}{cc} & {\dot{x}}_{0}\left(t\right)=v_{0}\left(t\right),\\ & {\dot{v}}_{0}\left(t\right)=u_{0}\left(t\right), \end{array}\right.$$
where $x_{0}\left(t\right)$∈ R and $v_{0}\left(t\right)$∈ R represent the states of the leader; $u_{0}\left(t\right)$ is the leader’s control input.

This paper aims to design a distributed adaptive event-triggered control protocol for each follower to achieve the following objectives:(1)All signals that are caused by closed-loop systems are bounded globally uniformly on [0, +∞);(2)There is no Zeno behavior;(3)The tracking errors, ${\bar{x}}_{i}\left(t\right)$ = $x_{i}\left(t\right)$−$x_{0}\left(t\right)$, ${\bar{v}}_{i}\left(t\right)$ = $v_{i}\left(t\right)$−$v_{0}\left(t\right)$, converge to a small adjustable bounded set.

In order to realize the control objectives, the following assumptions are imposed on systems (1) and (2)

**Assumption** **1.**
*A directed graph G contains a spanning tree whose leader is the root node, and the communication topology between all followers is undirected.*


**Assumption** **2.**
*The sign of $b_{i}$ is known.*


**Assumption** **3.**
*External disturbance $d_{i}$(t) is bounded: that is, $|d_{i}\left(t\right)|\leq \bar{d^{*}}$, where $\bar{d^{*}}$ is an unknown positive constant.*


**Assumption** **4.**
*The leader’s control input $u_{0}\left(t\right)$ is bounded: i.e., $|u_{0}\left(t\right)|\leq \rho$, where ρ is an unknown positive constant.*


## 3. Event-Based Distributed Adaptive Control Protocol Design

### 3.1. Triggering Mechanism

Event times for the *j*-th agent are denoted as: $t_{0}^{j},t_{1}^{j},\cdots ,t_{k}^{j},\cdots$ with 0 $\leq t_{0}^{j}<t_{1}^{j}<t_{2}^{j}<\cdots <t_{k}^{j}<t_{k+1}^{j}<+\infty$ for *k*∈ $Z^{+}$, *j* = 0, 1,⋯, *N*. The variable $t_{k}^{j}$ denotes the *k*-th event time for the *j*-th agent. In this paper, ${\overset{ˇ}{x}}_{j}$ and ${\overset{ˇ}{v}}_{j}$ are adopted to denote the sampling states for the *j*-th agent, and ${\overset{ˇ}{x}}_{j}\left(t\right)$ = $x_{j}\left(t_{k}^{j}\right)$, ${\overset{ˇ}{v}}_{j}\left(t\right)$ = $v_{j}\left(t_{k}^{j}\right)$, $t\in [t_{k}^{j},t_{k+1}^{j})$, which can be broadcast to neighbors at $t=t_{k}^{j}$. Meanwhile, the *j*-th agent updates the controller at $t=t_{k}^{j}$.

The event times are defined as
$$t_{k+1}^{j}\;=\inf\left\{t>t_{k}^{j}|\;p_{j}\left(t\right)\geq 0\right\},\;j=0,1,\cdots ,N,$$
where $t_{1}^{j}=0$, and the triggering mechanism is designed as
(3)$$p_{j}\left(t\right)=|x_{j}\left(t\right)-{\overset{ˇ}{x}}_{j}\left(t\right)|+|v_{j}\left(t\right)-{\overset{ˇ}{v}}_{j}\left(t\right)|-q_{j},$$
where $q_{j}$ is a positive constant to be designed. The triggering mechanism (3) means that: for agent *j*, once an event is triggered (i.e., $p_{j}\left(t_{k}^{j}\right)=0$), information transmission and control updating occur; after $t_{k}^{j}$, the triggering mechanism (3) is continuously evaluated until $p_{j}\left(t_{k+1}^{j}\right)=0$, which means that the (k+1)-th event is triggered. During any two consecutive events, if the neighbors of the *j*-th agent are triggered, the triggering mechanism (3) will not be updated. Since the states of the neighboring agents are not used in the triggering mechanism (3) for the *j*-th agent, the event times of the *j*-th agent are determined by themselves. It is noted that the continuous monitoring of the neighbors’ states is avoided since the triggering mechanism (3) only needs the current states of the *j*-th agent and the sampling states of the *j*-th agent.

### 3.2. Adaptive Control Protocol

In this section, we devise the adaptive control protocol for each follower based on the above triggering mechanism, and we analyze the existence and uniqueness of solutions that are caused by closed-loop systems.

Firstly, according to ${\bar{x}}_{i}\left(t\right)$ = $x_{i}\left(t\right)$−$x_{0}\left(t\right)$, ${\bar{v}}_{i}\left(t\right)$ = $v_{i}\left(t\right)$−$v_{0}\left(t\right)$, we deduce that
(4)$$\left\{\;\;\;\;\begin{array}{cc} & {\dot{\bar{x}}}_{i}\left(t\right)={\bar{v}}_{i}\left(t\right),\\ & {\dot{\bar{v}}}_{i}\left(t\right)=b_{i}u_{i}\left(t\right)+\theta_{i}f_{i}\left(x_{i},v_{i},t\right)+d_{i}\left(t\right)-u_{0}\left(t\right),\;i=1,\cdots ,N. \end{array}\right.$$

For System (4), the following adaptive control protocol is devised for the *i*-th follower
(5)$$\left\{\;\;\;\;\begin{array}{cc} & u_{i}={\hat{\beta }}_{i}\alpha_{i},\\ & \alpha_{i}=-{\hat{\theta }}_{i}f_{i}-l_{0}\left(\epsilon_{i1}+\epsilon_{i2}\right)+{\hat{u}}_{i,0}-{\hat{d}}_{i}, \end{array}\right.$$
where ${\hat{\beta }}_{i}$ and ${\hat{\theta }}_{i}$ are the estimates of $\beta_{i}$ = $\frac{1}{b_{i}}$ and $\theta_{i}$, respectively; ${\hat{d}}_{i}$ is the estimate of $d_{i}$; ${\hat{u}}_{i,0}$ is the estimate of $u_{0}$ by the *i*-th follower; $l_{0}$ is a positive parameter to be designed; $\epsilon_{i1}$ and $\epsilon_{i2}$ are relative errors on $t\in [t_{k},t_{k+1})$. The triggering mechanism is integrated into Control Protocol (5) by relative errors $\epsilon_{i1}$ and $\epsilon_{i2}$. In (5), an estimate-based method is used to estimate unknown parameters.

In (5), the estimates’ dynamics for the *i*-th follower are designed as
(6)$$\left\{\;\;\;\;\begin{array}{cc} & {\dot{\hat{\theta }}}_{i}=r_{i1}\left(e_{i1}+e_{i2}\right)f_{i}-r_{i1}k_{i1}\left({\hat{\theta }}_{i}-\theta_{i,0}\right),\\ & {\dot{\hat{\beta }}}_{i}=-r_{i2}\left(e_{i1}+e_{i2}\right)\mathrm{sign}\left(b_{i}\right)\alpha_{i}-r_{i2}k_{i2}\left({\hat{\beta }}_{i}-\beta_{i,0}\right),\\ & {\dot{\hat{u}}}_{i,0}=-r_{i3}\left(e_{i1}+e_{i2}\right)-r_{i3}k_{i3}\left({\hat{u}}_{i,0}-u_{0,0}\right),\\ & {\dot{\hat{d}}}_{i}=r_{i4}\left(e_{i1}+e_{i2}\right)-r_{i4}k_{i4}\left({\hat{d}}_{i}-d_{i,0}\right), \end{array}\right.$$
where sign(·) is the sign function; $r_{il}$ and $k_{il}$, *l* = 1, 2, 3, 4, are positive constants; $\theta_{i,0}$, $\beta_{i,0}$, $u_{0,0}$, $d_{i,0}$ are constants; $e_{i1}$ and $e_{i2}$ are relative errors.

In (6) and (7), the involved relative errors $e_{i1}$, $e_{i2}$, $\epsilon_{i1}$, and $\epsilon_{i2}$ are defined as
(7)$$\left\{\;\;\;\;\begin{array}{cc} & e_{i1}\left(t\right)=\sum_{j=1}^{N}a_{ij}\left(x_{i}\left(t\right)-x_{j}\left(t\right)\right)+\mu_{i}\left(x_{i}\left(t\right)-x_{0}\left(t\right)\right),\\ & e_{i2}\left(t\right)=\sum_{j=1}^{N}a_{ij}\left(v_{i}\left(t\right)-v_{j}\left(t\right)\right)+\mu_{i}\left(v_{i}\left(t\right)-v_{0}\left(t\right)\right),\\ & \epsilon_{i1}\left(t\right)=\sum_{j=1}^{N}a_{ij}\left(x_{i}\left(t\right)-{\overset{ˇ}{x}}_{j}\left(t\right)\right)+\mu_{i}\left(x_{i}\left(t\right)-{\overset{ˇ}{x}}_{0}\left(t\right)\right),\\ & \epsilon_{i2}\left(t\right)=\sum_{j=1}^{N}a_{ij}\left(v_{i}\left(t\right)-{\overset{ˇ}{v}}_{j}\left(t\right)\right)+\mu_{i}\left(v_{i}\left(t\right)-{\overset{ˇ}{v}}_{0}\left(t\right)\right). \end{array}\right.$$

Next, we analyze the existence and uniqueness of solutions that are caused by closed-loop systems, consisting of (3)–(6). Note that the right-hand sides of differential Equations (4)–(6) are locally Lipschitz in (${\bar{x}}_{i},{\bar{v}}_{i}$, ${\hat{\theta }}_{i},{\hat{\beta }}_{i},{\hat{u}}_{i,0},{\hat{d}}_{i}$) on R×R×R×R×R×R and continuous in *t* on $R_{+}$. Therefore, by Peano’s existence theorem and the extension theorem [28], a closed-loop system has a unique solution on its maximal existence interval [0, $t_{f}$) with 0<$t_{f}$≤+∞ for any initial conditions ($x_{i}\left(0\right),v_{i}\left(0\right)$, ${\hat{\theta }}_{i}\left(0\right),{\hat{\beta }}_{i}\left(0\right),{\hat{u}}_{i,0}\left(0\right),{\hat{d}}_{i}\left(0\right)$, $x_{0}$(0), $v_{0}$(0)) on R×R×R×R×R×R×R×R.

## 4. Main Results

In this section, we firstly construct a Lyapunov function. By Lemma 2, we prove the constructed Lyapunov function is positive definite. Then, we prove that all closed-loop system signals are bounded and exclude Zeno behavior on $\left[0,t_{f}\right)$ with $t_{f}<+\infty$. Furthermore, for all signals that cause the closed-loop system to be bounded, we deduce that $t_{f}=+\infty$. Finally, we prove that the devised adaptive event-triggered control protocol can realize bounded consensus tracking.

**Theorem** **1.**
*Consider Systems (1) and (2) under Assumptions 1–4. The designed distributed adaptive control protocol (5) with event-triggering mechanism (3) can guarantee: (i) all closed-loop system signals $\left(i.e.,x_{i},v_{i},u_{i},{\hat{\theta }}_{i},{\hat{\beta }}_{i},{\hat{u}}_{i,0},{\hat{d}}_{i}\right)$ are bounded on [0,+∞); (ii) Zeno behavior is excluded; (iii) for each follower, the tracking errors ${\bar{x}}_{i}\left(t\right)$ and ${\bar{v}}_{i}\left(t\right)$ converge to an adjustable bounded set.*


**Proof.** Firstly, we prove that all signals that are caused by closed-loop systems are bounded on [0, $t_{f}$) with $t_{f}<+\infty$. According to the estimation errors for unknown parameters and the unknown leader’s control input, a Lyapunov function is constructed as
(8)$$V\;=\frac{1}{2}\delta^{T}\Theta \;\delta +\frac{1}{2}\sum_{i=1}^{N}\frac{1}{r_{i1}}{{\tilde{\theta }}_{i}}^{2}+\frac{1}{2}\sum_{i=1}^{N}\frac{|b_{i}|}{r_{i2}}{\tilde{\beta }}_{i}^{2}+\frac{1}{2}\sum_{i=1}^{N}\frac{1}{r_{i3}}{\tilde{u}}_{i,0}^{2}+\frac{1}{2}\sum_{i=1}^{N}\frac{1}{r_{i4}}{\tilde{d}}_{i}^{2},$$
where Θ = $\left(\begin{array}{cc} 2W & W\\ W & W \end{array}\right)$, δ(t)=$\left[\delta_{1}\left(t\right),\cdots ,\delta_{N}\left(t\right)\right]$ with $\delta_{i}$ = ${\left[{\bar{x}}_{i},{\bar{v}}_{i}\right]}^{T}$, ${\tilde{\theta }}_{i}$ = $\theta_{i}$−${\hat{\theta }}_{i}$, ${\tilde{\beta }}_{i}$ = $\beta_{i}$−${\hat{\beta }}_{i}$, ${\tilde{u}}_{i,0}$ = $u_{0}$−${\hat{u}}_{i,0}$, ${\tilde{d}}_{i}$ = $d_{i}$−${\hat{d}}_{i}$. Under Assumption 1, *W* is symmetric, which means that the matrix Θ is symmetric, and we obtain that *W* is positive definite by Lemma 1. In Lemma 2, Θ can be regarded as *Q*; 2W can be regarded as $Q_{11}$; *W* can be regarded as $Q_{12}$ and $Q_{22}$. Then, we can deduce that Θ > 0 by noting 2W > 0, $W^{T}{\left(2W\right)}^{-1}W>0$. Thus, we obtain that *V* is positive definite.With this in hand, the derivative of *V* regarding time can be computed as
(9)$$\dot{V}\;=\delta^{T}\Theta \;\dot{\delta }-\sum_{i=1}^{N}\frac{1}{r_{i1}}{\tilde{\theta }}_{i}{\dot{\hat{\theta }}}_{i}-\sum_{i=1}^{N}\frac{|b_{i}|}{r_{i2}}{\tilde{\beta }}_{i}{\dot{\hat{\beta }}}_{i}-\sum_{i=1}^{N}\frac{1}{r_{i3}}{\tilde{u}}_{i,0}{\dot{\hat{u}}}_{i,0}-\sum_{i=1}^{N}\frac{1}{r_{i4}}{\tilde{d}}_{i}{\dot{\hat{d}}}_{i}.$$From (4), it follows that
(10)$$\dot{\delta }\left(t\right)=\left(\begin{array}{cc} 0 & I_{N}\\ 0 & 0 \end{array}\right)\delta \left(t\right)+\left(\begin{array}{c} 0\\ Bu(t) \end{array}\right)+\left(\begin{array}{c} 0\\ \theta F(x,v,t) \end{array}\right)+\left(\begin{array}{c} 0\\ D(t) \end{array}\right)-\left(\begin{array}{c} 0\\ 1_{N}u_{0}\left(t\right) \end{array}\right),$$
where $B=\mathrm{diag}\left\{b_{1},\cdots ,b_{N}\right\},u\left(t\right)={\left[u_{1},\cdots ,u_{N}\right]}^{T},\theta =\mathrm{diag}\left\{\theta_{1},\cdots ,\theta_{N}\right\}$, $F\left(x,v,t\right)={\left[f_{1}\left(x_{1},v_{1},t\right),\cdots ,f_{N}\left(x_{N},v_{N},t\right)\right]}^{T}$, $D\left(t\right)={\left[d_{1}\left(t\right),\cdots ,d_{N}\left(t\right)\right]}^{T}.$Based on (5) and (10), (9) can be further computed as
$$\begin{array}{cc} \dot{V}\;= & \;\delta^{T}\left(\begin{array}{cc} 0 & 2W\\ 0 & W \end{array}\right)\delta +\sum_{i=1}^{N}\frac{1}{r_{i1}}{\tilde{\theta }}_{i}\left(r_{i1}\left(e_{i1}+e_{i2}\right)f_{i}-{\dot{\hat{\theta }}}_{i}\right)+\sum_{i=1}^{N}\frac{1}{r_{i3}}{\tilde{u}}_{i,0}(-r_{i3}\left(e_{i1}+e_{i2}\right)\\ \; & -{\dot{\hat{u}}}_{i,0})+\sum_{i=1}^{N}\frac{1}{r_{i4}}{\tilde{d}}_{i}\left(r_{i4}\left(e_{i1}+e_{i2}\right)-{\dot{\hat{d}}}_{i}\right)-\sum_{i=1}^{N}l_{0}\left(e_{i1}+e_{i2}\right)\left(\epsilon_{i1}+\epsilon_{i2}\right)\\ \; & +\sum_{i=1}^{N}\frac{|b_{i}|}{r_{i2}}{\tilde{\beta }}_{i}\left(-r_{i2}\left(e_{i1}+e_{i2}\right)\mathrm{sign}\left(b_{i}\right)\alpha_{i}-{\dot{\hat{\beta }}}_{i}\right). \end{array}$$Noting that $\delta^{T}\left(\begin{array}{cc} 0 & W\\ W & W \end{array}\right)\delta$ = $\delta^{T}\left(\begin{array}{cc} 0 & 2W\\ 0 & W \end{array}\right)\delta$, we have
(11)$$\begin{array}{cc} \dot{V}\;= & \;\delta^{T}\left(\begin{array}{cc} 0 & W\\ W & W \end{array}\right)\delta +\sum_{i=1}^{N}\frac{1}{r_{i1}}{\tilde{\theta }}_{i}\left(r_{i1}\left(e_{i1}+e_{i2}\right)f_{i}-{\dot{\hat{\theta }}}_{i}\right)+\sum_{i=1}^{N}\frac{1}{r_{i3}}{\tilde{u}}_{i,0}(-r_{i3}\left(e_{i1}+e_{i2}\right)\\ \; & -{\dot{\hat{u}}}_{i,0})+\sum_{i=1}^{N}\frac{1}{r_{i4}}{\tilde{d}}_{i}\left(r_{i4}\left(e_{i1}+e_{i2}\right)-{\dot{\hat{d}}}_{i}\right)-\sum_{i=1}^{N}l_{0}\left(e_{i1}+e_{i2}\right)\left(\epsilon_{i1}+\epsilon_{i2}\right)\\ \; & +\sum_{i=1}^{N}\frac{|b_{i}|}{r_{i2}}{\tilde{\beta }}_{i}\left(-r_{i2}\left(e_{i1}+e_{i2}\right)\mathrm{sign}\left(b_{i}\right)\alpha_{i}-{\dot{\hat{\beta }}}_{i}\right). \end{array}$$Substituting (6) into (11) yields
(12)$$\begin{array}{cc} \dot{V}\;= & \;\delta^{T}\left(\begin{array}{cc} 0 & W\\ W & W \end{array}\right)\delta +\;\sum_{i=1}^{N}k_{i1}{\tilde{\theta }}_{i}\left({\hat{\theta }}_{i}-\theta_{i,0}\right)+\;\sum_{i=1}^{N}k_{i2}\left|b_{i}\right|{\tilde{\beta }}_{i}\left({\hat{\beta }}_{i}-\beta_{i,0}\right)+\;\sum_{i=1}^{N}k_{i4}{\tilde{d}}_{i}\left({\hat{d}}_{i}-d_{i,0}\right)\\ \; & +\sum_{i=1}^{N}k_{i3}{\tilde{u}}_{i,0}\left({\hat{u}}_{i,0}-u_{0,0}\right)-\sum_{i=1}^{N}l_{0}\left(e_{i1}+e_{i2}\right)\left(\epsilon_{i1}+\epsilon_{i2}\right). \end{array}$$By [24], the following inequalities are acquired:
(13)$$\left\{\;\;\;\;\begin{array}{cc} & {\tilde{\theta }}_{i}\left({\hat{\theta }}_{i}-\theta_{i,0}\right)\leq -\frac{1}{2}{\tilde{\theta }}_{i}^{2}+\frac{1}{2}{\left(\theta_{i}-\theta_{i,0}\right)}^{2},\\ & {\tilde{\beta }}_{i}\left({\hat{\beta }}_{i}-\beta_{i,0}\right)\leq -\frac{1}{2}{\tilde{\beta }}_{i}^{2}+\frac{1}{2}{\left(\beta_{i}-\beta_{i,0}\right)}^{2},\\ & {\tilde{d}}_{i}\left({\hat{d}}_{i}-d_{i,0}\right)\leq -\frac{1}{2}{\tilde{d}}_{i}^{2}+\frac{1}{2}{\left(d_{i}-d_{i,0}\right)}^{2},\\ & {\tilde{u}}_{i,0}\left({\hat{u}}_{i,0}-u_{0,0}\right)\leq -\frac{1}{2}{\tilde{u}}_{i,0}^{2}+\frac{1}{2}{\left(u_{i,0}-u_{0,0}\right)}^{2}. \end{array}\right.$$Substituting (13) into (12), it can be deduced that
(14)$$\begin{array}{cc} \dot{V}\;\leq & \;\delta^{T}\left(\begin{array}{cc} 0 & W\\ W & W \end{array}\right)\delta -\sum_{i=1}^{N}\frac{k_{i1}}{2}{\tilde{\theta }}_{i}^{2}-\sum_{i=1}^{N}\frac{k_{i2}}{2}\left|b_{i}\right|{\tilde{\beta }}_{i}^{2}-\sum_{i=1}^{N}\frac{k_{i3}}{2}{\tilde{u}}_{i,0}^{2}-\sum_{i=1}^{N}\frac{k_{i4}}{2}{\tilde{d}}_{i}^{2}\\ \; & +\sum_{i=1}^{N}\frac{k_{i1}}{2}{\left(\theta_{i}-\theta_{i,0}\right)}^{2}+\sum_{i=1}^{N}\frac{k_{i2}}{2}\left|b_{i}\right|{\left(\beta_{i}-\beta_{i,0}\right)}^{2}+\sum_{i=1}^{N}\frac{k_{i3}}{2}{\left(u_{0}-u_{0,0}\right)}^{2}\\ \; & +\sum_{i=1}^{N}\frac{k_{i4}}{2}{\left(d_{i}-d_{i,0}\right)}^{2}-\sum_{i=1}^{N}l_{0}\left(e_{i1}+e_{i2}\right)\left(\epsilon_{i1}+\epsilon_{i2}\right). \end{array}$$ where $\epsilon_{i1}+\epsilon_{i2}=e_{i1}+e_{i2}+\sum_{j=1}^{N}a_{ij}\left(x_{j}\left(t\right)-{\overset{ˇ}{x}}_{j}\left(t\right)\right)+\mu_{i}\left(x_{0}\left(t\right)-{\overset{ˇ}{x}}_{0}\left(t\right)\right)+\sum_{j=1}^{N}a_{ij}\left(v_{j}\left(t\right)-{\overset{ˇ}{v}}_{j}\left(t\right)\right)+\mu_{i}\left(v_{0}\left(t\right)-{\overset{ˇ}{v}}_{0}\left(t\right)\right)$.Moreover, by Young’s inequality, we have
$$\begin{array}{c} l_{0}q|e_{i1}+e_{i2}\left|\cdot \right|\Delta_{i}+\mu_{i}|\leq \frac{l_{0}}{2}{\left(e_{i1}+e_{i2}\right)}^{2}+\frac{l_{0}}{2}q^{2}{\left(\Delta_{i}+\mu_{i}\right)}^{2}, \end{array}$$
where *q* = max{$q_{j}$, j=0,1,⋯,N}.By (7), note $\sum_{i=1}^{N}{\left(e_{i1}+e_{i2}\right)}^{2}=\delta^{T}{\left(\begin{array}{cc} W & 0\\ 0 & W \end{array}\right)}^{T}\left(\begin{array}{cc} W & 0\\ 0 & W \end{array}\right)\delta$. Then, from (14), it follows that
(15)$$\begin{array}{cc} \dot{V}\;\leq & -\delta^{T}\left(\frac{l_{0}}{2}{\left(\begin{array}{cc} W & 0\\ 0 & W \end{array}\right)}^{T}\left(\begin{array}{cc} W & 0\\ 0 & W \end{array}\right)-\left(\begin{array}{cc} 0 & W\\ W & W \end{array}\right)\right)\delta -\sum_{i=1}^{N}(\frac{k_{i1}}{2}{\tilde{\theta }}_{i}^{2}+\frac{k_{i2}}{2}\left|b_{i}\right|{\tilde{\beta }}_{i}^{2}\\ \; & +\frac{k_{i3}}{2}{\tilde{u}}_{i,0}^{2}+\frac{k_{i4}}{2}{\tilde{d}}_{i}^{2})+\sum_{i=1}^{N}(\frac{l_{0}}{2}q^{2}{\left(\Delta_{i}+\mu_{i}\right)}^{2}+\frac{k_{i1}}{2}{\left(\theta_{i}-\theta_{i,0}\right)}^{2}\\ \; & +\frac{k_{i2}}{2}\left|b_{i}\right|{\left(\beta_{i}-\beta_{i,0}\right)}^{2}+\frac{k_{i3}}{2}{\left(u_{0}-u_{0,0}\right)}^{2}+\frac{k_{i4}}{2}{\left(d_{i}-d_{i,0}\right)}^{2}). \end{array}$$Let Γ = ${\left(\begin{array}{cc} W & 0\\ 0 & W \end{array}\right)}^{T}$$\left(\begin{array}{cc} W & 0\\ 0 & W \end{array}\right)$, ∏ = $\left(\begin{array}{cc} 0 & W\\ W & W \end{array}\right)$. Based on (15), we obtain
(16)$$\begin{array}{cc} \dot{V}\;\leq & -\left(\frac{l_{0}}{2}\lambda_{\min}\left(\Gamma \right)-\lambda_{\max}\left(\prod \right)\right)\delta^{T}\delta -\sum_{i=1}^{N}\omega \left(\frac{{\tilde{\theta }}_{i}^{2}}{2r_{i1}}+\frac{|b_{i}|{\tilde{\beta }}_{i}^{2}}{2r_{i2}}+\frac{{\tilde{u}}_{i,0}^{2}}{2r_{i3}}+\frac{{\tilde{d}}_{i}^{2}}{2r_{i4}}\right)\\ \; & +\sum_{i=1}^{N}(\frac{l_{0}}{2}m^{2}{\left(\Delta_{i}+\mu_{i}\right)}^{2}+\frac{k_{i1}}{2}{\left(\theta_{i}-\theta_{i,0}\right)}^{2}+\frac{k_{i2}}{2}\left|b_{i}\right|{\left(\beta_{i}-\beta_{i,0}\right)}^{2}+\frac{k_{i3}}{2}{\left(u_{0}\;-\;u_{0,0}\right)}^{2}\\ \; & +\frac{k_{i4}}{2}{\left(d_{i}-d_{i,0}\right)}^{2}), \end{array}$$
where ω = min$\left\{k_{i1}r_{i1},k_{i2}r_{i2},k_{i3}r_{i3},k_{i4}r_{i4}\right\}$.Furthermore, from (8), it follows that
(17)$$\begin{array}{c} V\;\leq \frac{1}{2}\lambda_{\max}\left(\Theta \right)\delta^{T}\delta +\sum_{i=1}^{N}\left(\frac{{\tilde{\theta }}_{i}^{2}}{2r_{i1}}+\frac{|b_{i}|{\tilde{\beta }}_{i}^{2}}{2r_{i2}}+\frac{{\tilde{u}}_{i,0}^{2}}{2r_{i3}}+\frac{{\tilde{d}}_{i}^{2}}{2r_{i4}}\right). \end{array}$$Combining (16) with (17), we obtain
$$\begin{array}{c} \dot{V}\;\leq -\left(\frac{l_{0}}{2}\lambda_{\min}\left(\Gamma \right)-\lambda_{\max}\left(\prod \right)-\frac{\omega }{2}\lambda_{\max}\left(\Theta \right)\right)\delta^{T}\delta -\omega V+M^{*}, \end{array}$$
where $M^{*}$ = $\sum_{i=1}^{N}$($\frac{l_{0}}{2}q^{2}$${\left(\Delta_{i}+\mu_{i}\right)}^{2}$ + $\frac{k_{i1}}{2}$${\left(\theta_{i}-\theta_{i,0}\right)}^{2}$ + $\frac{k_{i2}}{2}\left|b_{i}\right|$${\left(\beta_{i}-\beta_{i,0}\right)}^{2}$ + $\frac{k_{i3}}{2}$${\left(u_{0}-u_{0,0}\right)}^{2}$ + $\frac{k_{i4}}{2}$${\left(d_{i}-d_{i,0}\right)}^{2}$). Then, by choosing $l_{0}$>$\lambda_{\min}^{-1}$(Γ)($\omega \lambda_{\max}\left(\Theta \right)+2\lambda_{\max}\left(\prod \right)$), we deduce that
(18)$$\begin{array}{c} \dot{V}\;\leq -\omega V+M^{*}. \end{array}$$By solving (18), we derive
(19)$$\begin{array}{c} V\left(t\right)\;\leq V\left(0\right)e^{-\omega t}+\frac{M^{*}}{\omega }\left(1-e^{-\omega t}\right), \end{array}$$
which shows that V(t) is uniformly bounded on [0, $t_{f}$). Thus, the signals $\delta_{i}$, ${\hat{\theta }}_{i}$, ${\hat{\beta }}_{i}$, ${\hat{u}}_{i,0}$, and ${\hat{d}}_{i}$ are bounded on [0, $t_{f}$). Under Assumption 4, it is concluded that $x_{0}$ and $v_{0}$ are bounded on [0, $t_{f}$). Therefore, $x_{i}$ and $v_{i}$ are bounded by the boundedness of $\delta_{i}$, $x_{0}$, $v_{0}$ on [0, $t_{f}$). Furthermore, $u_{i}$ is bounded on [0, $t_{f}$) from (5).Now, we prove that there is no Zeno behavior on [0, $t_{f}$): that is, the lower bound on the interval between two consecutive events {$t_{k+1}^{j}-$$t_{k}^{j}$} is positive.Define $\varphi_{k}^{j}\left(t\right)$ = $|x_{j}\left(t\right)$−${\overset{ˇ}{x}}_{j}\left(t\right)|$ + $|v_{j}\left(t\right)$−${\overset{ˇ}{v}}_{j}\left(t\right)|$ for *t*∈ $\left[t_{k}^{j},t_{k+1}^{j}\right)$. First, the derivative of the triggering mechanism $\varphi_{k}^{j}\left(t\right)$ regarding time needs to be acquired. Then, the maximum velocity $h_{j}$ of the change in $\varphi_{k}^{j}\left(t\right)$ from 0 to $q_{j}$ can be acquired. We exclude Zeno behavior by testing if $\frac{q_{j}}{h_{j}}$ is positive. The derivative of $\varphi_{k}^{j}\left(t\right)$ regarding time is computed as
$$\begin{array}{cc} {\dot{\varphi }}_{k}^{j}\left(t\right) & =D^{+}|x_{j}\left(t\right)|+D^{+}\left|v_{j}\left(t\right)\right|\\ \; & =|v_{j}\left(t\right)|+|b_{j}u_{j}\left(t\right)+f_{j}\left(x_{j},v_{j},t\right)\theta_{j}+d_{j}\left(t\right)|,\;j=0,1,\cdots ,N \end{array}$$
and ${\dot{\varphi }}_{k}^{0}\left(t\right)$≤∣${\dot{x}}_{0}$(t)∣+∣${\dot{v}}_{0}$(t)∣. By the boundedness of $u_{j}$, $v_{j}$, ${\dot{x}}_{0}$, and ${\dot{v}}_{0}$ on [0, $t_{f}$), we acquire a positive number $h_{j}$ such that ${\dot{\varphi }}_{k}^{j}\left(t\right)$≤$h_{j}$. Moreover, we get that $\int_{t_{k+1}^{j}}^{t_{k}^{j}}{\dot{\varphi }}_{k}^{j}\left(t\right)dt\leq \int_{t_{k+1}^{j}}^{t_{k}^{j}}h_{j}dt\;=\;h_{j}\left(t_{k+1}^{j}-t_{k}^{j}\right)$. From (3), it follows that $\varphi_{k+1}^{j}\left(t_{k+1}^{j}\right)-\varphi_{k}^{j}\left(t_{k}^{j}\right)>q_{j}$. Thus, we arrive at $t_{k+1}^{j}$−$t_{k}^{j}$≥$\frac{q_{j}}{h_{j}}$. Zeno behavior is excluded on [0, $t_{f}$). Furthermore, all signals that cause closed-loop systems are bounded; we deduce that $t_{f}$ = +*∞*.Finally, that the tracking errors ${\bar{x}}_{i}\left(t\right)$ and ${\bar{v}}_{i}\left(t\right)$ converge to a small adjustable bounded set was testified. From (9), it follows that
$$V\left(t\right)\geq \frac{1}{2}\delta^{T}\left(t\right)\left(\begin{array}{cc} 2W & W\\ W & W \end{array}\right)\delta \left(t\right)\geq \frac{1}{2}\lambda_{\min}\left(\Theta \right)\;{\left\|\delta \left(t\right)\right\|}^{2},$$
then,
(20)$${\left\|\delta \left(t\right)\right\|}^{2}\leq \frac{2}{\lambda_{\min}\left(\Theta \right)}V\left(t\right).$$Substituting (19) into (20), we obtain
$${\left\|\delta \left(t\right)\right\|}^{2}\leq \frac{2}{\lambda_{\min}\left(\Theta \right)}\left[V\left(0\right)e^{-\omega t}+\frac{M^{*}}{\omega }\left(1-e^{-\omega t}\right)\right],$$
which implies that ${\left\|\delta \left(t\right)\right\|}^{2}$ is bounded and converges to a compact set $E_{r}$ = {δ(t) | ${\left\|\delta \left(t\right)\right\|}^{2}$ ≤ $\frac{2}{\omega \lambda_{\min}\left(\Theta \right)}$ ($M^{*}$+σ)} for *t* ≥ (1/ω) ln(∣ωV(0)−$M^{*}$∣/σ) with σ an arbitrarily small positive constant. Note ${\left\|\delta \left(t\right)\right\|}^{2}={\bar{x}}_{1}^{2}+\cdots +{\bar{x}}_{N}^{2}+{\bar{v}}_{1}^{2}+\cdots +{\bar{v}}_{N}^{2}$. Therefore, tracking errors ${\bar{x}}_{i}$(t) and ${\bar{v}}_{i}$(t) also converge to $E_{r}$ for *t*≥ (1/ω) ln(∣ωV(0)−$M^{*}$∣/σ). It is noted that the compact set $E_{r}$ can be made as small as desired by increasing $r_{i1},r_{i2},r_{i3},r_{i4}$ while fixing all the remaining design parameters. Thus, the compact set $E_{r}$ can be adjusted by modifying the relevant parameters. □

## 5. Simulation of Nonlinear MASs with Uncertainties

In this section, we provide a simulation example in detail to verify the theoretical results obtained. Consider an uncertain MAS with four followers and one leader. The *i*-th follower’s dynamics, i=1,⋯,4, are described as (1), where $b_{1}$ = $b_{2}$ = $b_{3}$ = 1, $b_{4}$ = 1.1, $\theta_{1}$ = 1, $\theta_{2}$ = 2, $\theta_{3}$ = 3, $\theta_{4}$ = 4, $f_{1}\left(x_{1},v_{1},t\right)=0.1x_{1}^{2}+0.1v_{1}^{2}$, $f_{2}\left(x_{2},v_{2},t\right)=0.2x_{2}^{2}+0.1v_{2}^{3}$, $f_{3}\left(x_{3},v_{3},t\right)=0.2x_{3}^{2}+0.1v_{3}^{2}$, $f_{4}\left(x_{4},v_{4},t\right)=0.1x_{4}^{3}+0.1v_{4}^{2}$, $d_{1}\left(t\right)=\sin\left(t\right),d_{2}\left(t\right)=1.1\sin\left(t\right),d_{3}\left(t\right)=1.2\sin\left(t\right),d_{4}\left(t\right)=2\sin\left(t\right)$. The leader’s dynamics are described by (2), where $u_{0}\left(t\right)=-\sin\left(t\right)$.

Note that the communication topology for systems (21) and (22) is shown in Figure 1. The four followers’ initial conditions and the leader’s conditions are given as follows: $x_{1}\left(0\right)$ = 0, $v_{1}\left(0\right)$ = 1, $x_{2}\left(0\right)$ = 1, $v_{2}\left(0\right)$ = 2, $x_{3}\left(0\right)$ = 2, $v_{3}\left(0\right)$ = 3, $x_{4}\left(0\right)$ = 4, $v_{4}\left(0\right)$ = 2, $x_{0}\left(0\right)$ = 0, $v_{0}\left(0\right)$ = 1. The initial conditions ${\hat{\theta }}_{i}\left(0\right)$, ${\hat{\beta }}_{i}\left(0\right)$, ${\hat{u}}_{i,0}\left(0\right)$, ${\hat{d}}_{i}\left(0\right)$, i=1,⋯,4 are set to zero. The corresponding parameters in (6) and (7) are chosen as $l_{0}$ = 170, $k_{i1}$ = $k_{i2}$ = $k_{i3}$ = $k_{i4}$ = 5, $r_{i1}$ = $r_{i2}$ = $r_{i3}$ = $r_{i4}$ = 0.005, $\theta_{i,0}$ = $\beta_{i,0}$ = $u_{0,0}$ = $d_{i,0}$ = 0. The designed parameters in (3) are taken as $m_{1}=0.95,m_{2}=1,m_{3}=2,m_{4}=2.65,m_{0}=1.1$. From Figure 2, we can see that the evolution of tracking errors, ${\bar{x}}_{i}=x_{i}-x_{0}$, ${\bar{v}}_{i}=v_{i}-v_{0}$, converges to a bounded set. Figure 3 illustrates that the evolution of the control protocols for the four followers is bounded. The four followers’ triggering times and the leader’s triggering times are exhibited in Figure 4.

In light of the work [27], a distributed adaptive control protocol without ETC is devised to achieve consensus tracking for systems (1) and (2). The trajectories of tracking errors, ${\bar{x}}_{i}$ and ${\bar{v}}_{i}$, are shown in Figure 5.

The simulation experiments are run to make Table 2, which shows the average consensus time (ACT) and the integral of the absolute value of the input (IAU). ACT means the average time it takes for followers to track the leader’s trajectory. IAU can evaluate the energy consumption of control signals [29], where larger values of IAU mean greater energy consumption for the control signals.

From Table 2, we acquire that the devised consensus protocol in this paper can make all followers track the leader faster. Further, when the triggering mechanism is considered in the design of the control protocol, energy resources can be better conserved.

## 6. Conclusions

This paper has addressed bounded consensus tracking for nonlinear MASs with uncertainties by ETC. The considered MASs permit multiple uncertainties, including an unknown control coefficient, parameterized unknown nonlinearities, unknown external disturbances, and unknown control input of the leader. To compensate for uncertainties, a new estimate-based adaptive control protocol has been proposed. Integrating a triggering mechanism into the adaptive control protocol, the bounded consensus tracking of MASs can be achieved with fewer communication resources. Then, by choosing an appropriate Lyapunov function candidate, it has been proved that the designed control protocol can guarantee that there is no Zeno behavior and that tracking errors ultimately converge to a bounded set. For future study, extending the results to finite-time consensus will be interesting work.

## Figures and Tables

**Figure 1 entropy-25-01335-f001:**
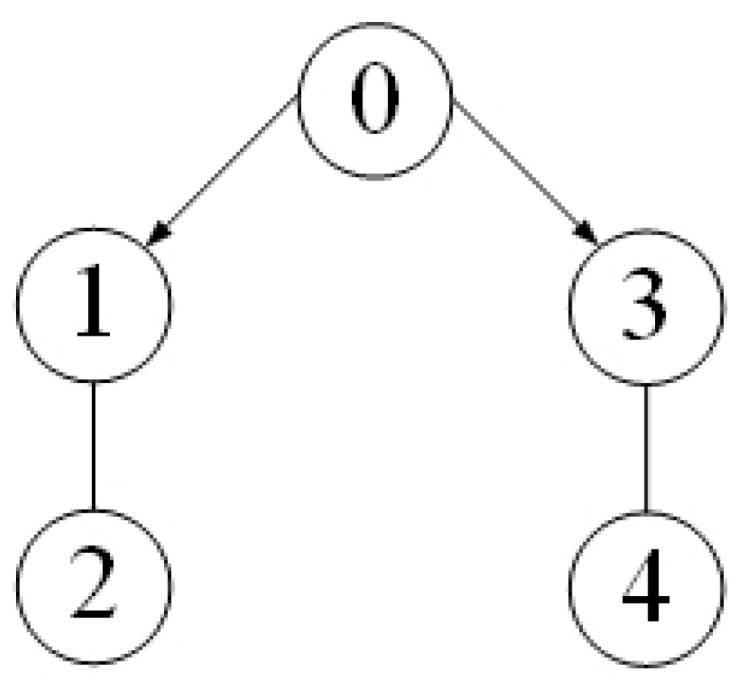
Communication topology among a leader and all followers.

**Figure 2 entropy-25-01335-f002:**
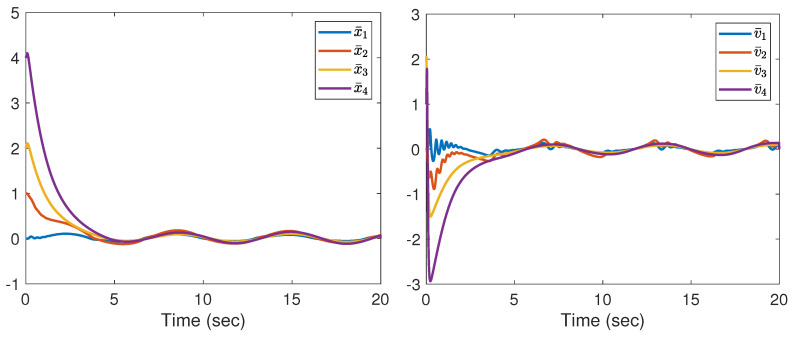
The evolution of ${\bar{x}}_{i}$ and ${\bar{v}}_{i}$.

**Figure 3 entropy-25-01335-f003:**
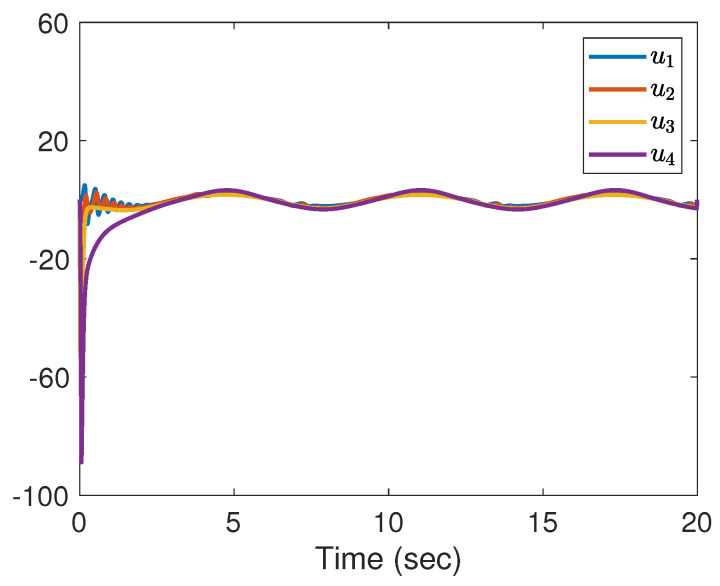
The evolution of control protocol for followers.

**Figure 4 entropy-25-01335-f004:**
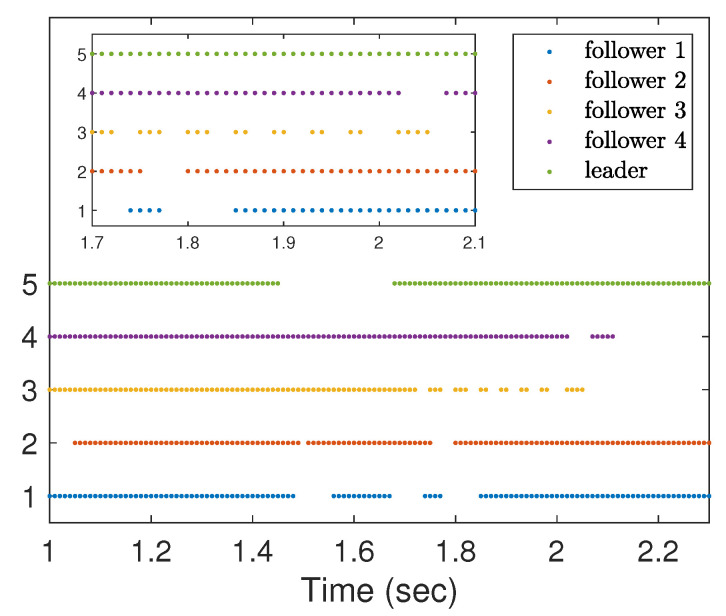
Triggering times of four followers and one leader.

**Figure 5 entropy-25-01335-f005:**
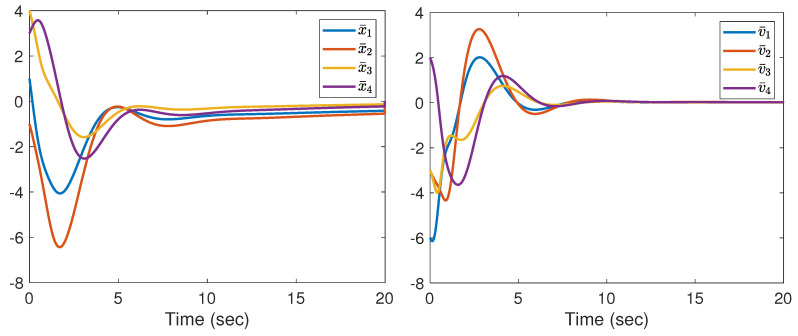
The evolution of tracking errors ${\bar{x}}_{i}$ and ${\bar{v}}_{i}$.

**Table 1 entropy-25-01335-t001:** Comparison of the features of the investigated MASs in the existing literature.

Work	Control Coefficient	Lipschitz Constant	External Disturbance	Leader Input
[21]	No	Known	No	Known
[22]	No	Unknown	No	Zero
[22]	No	Unknown	No	Zero
[23]	Yes	Unknown	No	Unknown
this paper	Yes	Unknown	Yes	Unknown

**Table 2 entropy-25-01335-t002:** Comparison between the consensus control protocol with triggering mechanism (CWT) and the consensus control protocol without triggering mechanism (CWNT).

Protocol	ACT	IAU
CWT	5 s	205.4932
CWNT	8.5 s	622.5718

## Data Availability

Data is unavailable due to privacy.

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
