# Peer review of "Event-Triggered Bounded Consensus Tracking for Second-Order Nonlinear Multi-Agent Systems with Uncertainties"

_entropy, 2023, doi:10.3390/e25091335_

Round 1
Reviewer 1 Report
This paper aims at presenting an event-triggered adaptive control method for nonlinear multi agent systems with uncertainties.
The motivation and practical relevance of the proposed work is not clear. Although a discussion of the literature is proposed, it lacks in properly clarifying the positioning of the paper with respect to the existing works.
Then, a more detailed discussion of the motivation of this work must be provided and the analysis of the state of the art must be extended. It would be helpful the inclusion of a table to compare the contribution of this paper with respect to existing works. It would be also beneficial the inclusion of a literature discussion regarding the control of hybrid systems (see e.g. 10.1109/LRA.2015.2502905, 10.1109/TAC.2020.2979788)
Section 2 should be integrated with preliminaries on adaptive control and event-triggered control. In fact, the proposed method is a control strategy for hybrid systems, while no discussion is provided in such regards.
Section 3 - The triggering mechanism is barely discussed. Which is the event that triggers the broadcast of the information among the agents? Eq(3) must be discussed more in detail. Also the parameters in eqs(6)-(7) have to be motivated and their setting has to be explained.
Section 4 - The example should highlight the advantages of the proposed method with respect to other existing techniques, thus a comparison with other literature methodologies has to be included.
Minor typos are present in the paper that need corrections.
Reviewer 2 Report
This paper is concerned with event-triggered bounded consensus tracking for a class of second-order uncertain nonlinear multi-agent systems. This paper is well written and contains some interesting results. The following comments are provided to help further improve the paper.
1) Assumption 2 requires to know the sign of b_i, which may be a little restrictive. Is that possible to consider a more relaxed assumption? Such as unknown sign of b_i?
2) The triggering mechanism (3) looks to require continuous monitoring neighbor states. If so, how to solve this problem. Can you redesign a new controller to overcome it?
3) Please confirm the expression of (11) and (13), which are key for the whole paper. Particularly, the one in (11) looks difficult to obtain.
4) In simulation, the results seems good. But my concern is Figure 4, it looks like Follower 3 and Follower 4 keeps triggering? Please provide more detail to explain it? How about Zeno?
5) This paper is concerned with cooperative control of MASs, whose application on mobile robots may deserve authors' attention such as Collective behaviors of mobile robots beyond the nearest neighbor rules with switching topology.
6) The conclusion section can be enhanced by discussing some future directions such as extending the asymptotic convergence to the fixed-time or prescribed-time convergence. The authors may refer to some latest survey paper in this area.
The English language can be further improved.
Round 2
Reviewer 1 Report
The paper has been sufficiently revised by the authors and the content as well as the contributions of the paper are more clear.